# Endosomal trafficking of two-pore K⁺ efflux channel TWIK2 to plasmalemma mediates NLRP3 inflammasome activation and inflammatory injury

**Long Shuang Huang[1,2], Mohammad Anas[1], Jingsong Xu[1], Bisheng Zhou[1], Peter T Toth[3], Yamuna Krishnan[4], Anke Di[1]\*, Asrar B Malik[1]\***

[1]Department of Pharmacology and Regenerative Medicine, The University of Illinois College of Medicine, Chicago, United States; [2]Shanghai Frontiers Science Center of Drug Target Identification and Delivery, School of Pharmacy, Shanghai Jiao Tong University, Shanghai, China; [3]Fluorescence Imaging Core, The University of Illinois College of Medicine, Chicago, United States; [4]Department of Chemistry, University of Chicago, Chicago, United States

**\*For correspondence:**
ankedi@uic.edu (AD);
abmalik@uic.edu (ABM)

**Competing interest:** The authors declare that no competing interests exist.

**Abstract** Potassium efflux via the two-pore K⁺ channel TWIK2 is a requisite step for the activation of NLRP3 inflammasome, however, it remains unclear how K⁺ efflux is activated in response to select cues. Here, we report that during homeostasis, TWIK2 resides in endosomal compartments. TWIK2 is transported by endosomal fusion to the plasmalemma in response to increased extracellular ATP resulting in the extrusion of K⁺. We showed that ATP-induced endosomal TWIK2 plasmalemma translocation is regulated by Rab11a. Deleting Rab11a or ATP-ligated purinergic receptor P2X7 each prevented endosomal fusion with the plasmalemma and K⁺ efflux as well as NLRP3 inflammasome activation in macrophages. Adoptive transfer of Rab11a-depleted macrophages into mouse lungs prevented NLRP3 inflammasome activation and inflammatory lung injury. We conclude that Rab11a-mediated endosomal trafficking in macrophages thus regulates TWIK2 localization and activity at the cell surface and the downstream activation of the NLRP3 inflammasome. Results show that endosomal trafficking of TWIK2 to the plasmalemma is a potential therapeutic target in acute or chronic inflammatory states.

## Editor's evaluation

This important study focuses on the mechanism by which potassium channels are activated prior to NLRP3 inflammasome activation. Using convincing microscopy and biochemical studies, the authors demonstrate that the potassium channel, TWIK2, located in the endosomal compartment during basal conditions, is translocated onto the plasmalemma upon ATP stimulation, which triggers potassium efflux and subsequent NLRP3 inflammasome activation. These findings address a long-standing question in the inflammasome field.

## Introduction

Inflammasomes are key components of the immune system and inflammatory signaling platforms responsible for detecting injury mediators released during infection and tissue damage and that thereby initiate the inflammatory response (*Davis et al., 2011*; *Broz and Dixit, 2016*; *Sharma and Kanneganti, 2021*; *Sefik et al., 2022*). NLRP3 (Nucleotide-binding oligomerization domain-Like

Receptor containing Pyrin domain 3) inflammasome expressed in immune cells such as macrophages is a key determinant of acute immune responses such as acute lung injury and COVID-19 (*Sefik et al., 2022*; *Swanson et al., 2019*; *Grailer et al., 2014*) as well as chronic inflammatory diseases such as atherosclerosis, cancer, or metabolic syndrome (*Sharma and Kanneganti, 2021*; *Baldrighi et al., 2017*). Activation of the NLRP3 inflammasome complex is a multi-step process involving the assembly of essential proteins, and activation of caspase 1 which cleaves pro-interleukin-1β to release the active form of this inflammatory cytokine (*Swanson et al., 2019*; *He et al., 2016*). NLRP3 inflammasome consists of a sensor (NLRP3), an adaptor (apoptosis-associated speck-like protein containing a caspase recruitment domain – ASC), and an effector (caspase 1) (*Davis et al., 2011*; *Broz and Dixit, 2016*; *Sharma and Kanneganti, 2021*; *Sefik et al., 2022*). Oligomerized NLRP3 recruits ASC which in turn recruits caspase 1 and enables proximity-induced caspase 1 self-cleavage and activation (*Davis et al., 2011*; *Broz and Dixit, 2016*; *Sharma and Kanneganti, 2021*; *Sefik et al., 2022*). Several studies have elucidated the structure and assembly of the NLRP3 complex. The inactive NLRP3 form (double-ring cages of NLRP3) is membrane-associated organelles (such as endoplasmic reticulum [ER], mitochondria, and Golgi apparatus) and is recruited, assembled, and activated at the centrosome (*Andreeva et al., 2021*; *Zhou et al., 2011*; *Subramanian et al., 2013*; *Wang et al., 2020*; *Chen and Chen, 2018*; *Wu et al., 2022*; *Li et al., 2017*; *Yang et al., 2020*; *Magupalli et al., 2020*). However, little is known about the initiating triggers of the assembly and activation of NLRP3 complex.

We showed that essential mechanism of NLRP3 assembly involves the efflux of potassium ($K^+$) at the plasmalemma (*Pétrilli et al., 2007*; *Franchi et al., 2007*; *Muñoz-Planillo et al., 2013*) through the potassium channel TWIK2 (the Two-pore domain Weak Inwardly rectifying $K^±$ channel 2), a member of the two-pore domain $K^+$ channel ($K_{2P}$) family ($K_{2P}$ 6.1, encoded by *Kcnk6*) (*Enyedi and Czirják, 2010*; *Di et al., 2018*). Efflux of potassium generates regions of low intracellular $K^+$ which promote conformational change of the inactive NLRP3 to facilitate NLRP3 assembly and activation (*Tapia-Abellán et al., 2021*). Based on our model, TWIK2-mediated plasmalemmal potassium efflux thus serves as a checkpoint for the initiation of adaptive host defense signaling as well as maladaptive inflammatory signaling mediated by NLRP3 (*Di et al., 2018*). This essential function of TWIK2 raises the question how TWIK2 activity is fine tuned to avoid premature or intracellular triggering of NLRP3 during homeostasis while at the same time providing a rapid TWIK2 activation mechanism in response to extracellular tissue damage. Due to the high gradient of $K^+$ across the plasmalemma, the presence of TWIK2 at the plasma membrane results in basal $K^+$ efflux and thus may lead to inappropriate inflammasome activation. Therefore, the question of fine control of TWIK2-mediated $K^+$ efflux becomes prescient.

The activity of several ion channels such as the G-protein-activated inwardly rectifying $K^+$ (GIRK) channels (*Chung et al., 2009b*; *Chung et al., 2009a*; *Grant and Donaldson, 2009*) and cardiac pacemaker channels – hyperpolarization-activated cyclic nucleotide-gated (HCN) ion channels HCN2 and HCN4 is regulated by endosomal trafficking from the cytosol to the plasmalemma (*Hardel et al., 2008*). Here, we investigated whether TWIK2 can be sequestered in the steady state in cytosolic endosomal compartments to shield cells from pathogenic inflammasome activation through the efflux of $K^+$ and that TWIK2 is only trafficked to the plasmalemma on a need basis in response to cues elicited by tissue injury that trigger potassium efflux and activate NLRP3.

Using confocal and electron microscopy and electrophysiological studies, we found that TWIK2 $K^+$ channel in macrophages was expressed in endosomes at rest but translocated to the plasmalemma upon extracellular ATP challenge or release, an indicator of tissue damage. We demonstrated that the $Ca^{2+}$-sensitive GTP-binding protein Rab11a was responsible for endosomal TWIK2 translocation to the plasmalemma. Furthermore, inhibition of endosomal fusion with the plasma membrane prevented NLRP3 inflammasome activation. Adoptive transfer of Rab11-depleted macrophages into mouse lungs prevented NLRP3 inflammasome activation and inflammatory lung injury. The studies thus demonstrate a mechanism by which endosomal trafficking of TWIK2 and $K^+$ efflux trigger NLRP3 inflammasome activation without self-harm to the cell. Further the results point to inhibition of endosomal plasmalemma fusion as a potential anti-inflammatory therapy target.

## Results

### Endosomal TWIK2 plasmalemmal translocation induced by ATP in macrophages

TWIK2 belongs to the constitutively active $K_{2P}$ background potassium channel family (*Enyedi and Czirják, 2010*); however, TWIK2 plasmalemmal currents in macrophages are only observed following challenge with extracellular ATP (*Di et al., 2018*). A explanation for this funding is that TWIK2 is not present at the plasmalemma during the basal state and can only trafficked to the plasmalemma in response to environmental cues, similar to reports for activation of other ion channels (*Chung et al., 2009b*; *Chung et al., 2009a*; *Grant and Donaldson, 2009*; *Hardel et al., 2008*). To address this question, we imaged intracellular TWIK2 plasmalemma translocation upon ATP challenge by expressing TWIK2-GFP in macrophages. TWIK2-GFP plasmids were transfected into RAW 264.7 macrophages for 48 hr after which the cells were imaged with confocal microscopy after challenge with extracellular ATP to mimic tissue injury. TWIK2 translocated toward the plasmalemma within 2 min of extracellular ATP addition compared with the cells without the addition of extracellular ATP (*Figure 1A*, *Figure 1—video 1*, *Figure 1—video 2*). To assess TWIK2 plasmalemma insertion after ATP challenge, we examined the distribution of TWIK2 in cells before and after ATP challenge using confocal microscopy and immunogold-labeled electron microscopy in RAW 264.7 macrophages transfected with TWIK2-GFP plasmids (for confocal microscopy) or TWIK2 plasmids (for immunogold-labeled electron microscopy). Confocal imaging showed TWIK2 intracellular distribution before the ATP challenge, and clear evidence of TWIK2 plasmalemmal translocation after ATP challenge as evident using the plasma membrane marker (*Figure 1B*) and by immunogold-labeled electron microscopy (*Figure 1C*, *Figure 1—figure supplement 1*). Intracellular TWIK2 plasmalemmal translocation was also shown to be ATP concentration dependent (*Figure 1D*).

To identify the location of intracellular TWIK2, we labeled TWIK2 with fluorescent anti-TWIK2 antibody combined with fluorescence-labeled markers for early endosomes (EE) with EEA1 (Early Endosome Antigen-1) antibody, recycling endosomes (RE) with Rab11a antibody, lysosomes with LAMP1 (Lysosomal-Associated Membrane Protein 1) antibody, and ER with PDI (Protein Disulfide Isomerase) antibody in RAW 264.7 macrophages transfected with TWIK2 plasmids. We observed that TWIK2 was present primarily in endosomes (both EE and RE) and much less was seen in lysosomes or ER (*Figure 2*). Thus, TWIK2 distributed in endosomes during homeostasis and only translocated to the plasmalemmal upon ATP challenge. The weak staining of TWIK2 in lysosomes may be the result of background endosomal fusion with lysosomes (*Luzio et al., 2007*). The defective staining of TWIK2 seen in the ER reflects the role of ER in protein biosynthesis and the ER membrane as the site of production of transmembrane proteins (*Reid and Nicchitta, 2015*).

### ATP-induced exocytosis promotes plasmalemma potassium efflux and NLRP3 inflammasome activation in P2X7- and $Ca^{2+}$-dependent manner

Endosomal distribution of TWIK2 and plasmalemmal translocation suggest a mechanism of plasmalemma insertion of TWIK2 upon ATP challenge. To address TWIK2 localization, we monitored exocytosis (the fusion event of intracellular vesicles with plasmalemma) using membrane capacitance measurements that reflect increased membrane surface area due to vesicle–plasmalemma fusion (*Di et al., 2002*; *Di et al., 2001*). We observed ATP-induced exocytosis as reflected by increased plasmalemma capacitance measurements in monocyte-derived macrophages (MDMs) (*Figure 3A*). ATP-induced exocytosis determined by capacitance increases was inhibited by either deletion of P2X7 ($P2x7^{-/-}$), removal of extracellular $Ca^{2+}$, or using the vesicle–plasma fusion inhibitor (Vacuolin) *Ye et al., 2021*; *Cerny et al., 2004* in MDMs (*Figure 3A*).

We next investigated whether inhibiting ATP-induced vesicle–plasmalemma fusion or the reduction in extracellular $Ca^{2+}$ altered ATP-induced potassium current and NLRP3 inflammasome activation in MDMs. First, we assessed the role of the vesicle–plasmalemma fusion in ATP-induced current and NLRP3 inflammasome activation. MDMs pretreated with Vacuolin showed significantly reduced ATP-induced current (*Figure 3B*) and NLRP3 inflammasome activation (*Figure 3C–H*). NLRP3 inflammasome activation was evaluated by measuring caspase 1 activation (indicated by expression of p20 derived from pro-caspase 1), IL-1β maturation (indicated by p17 derived from pro-IL-1β), and the release of IL-1β and IL-18. We observed that caspase 1 activation, IL-1β maturation and the release

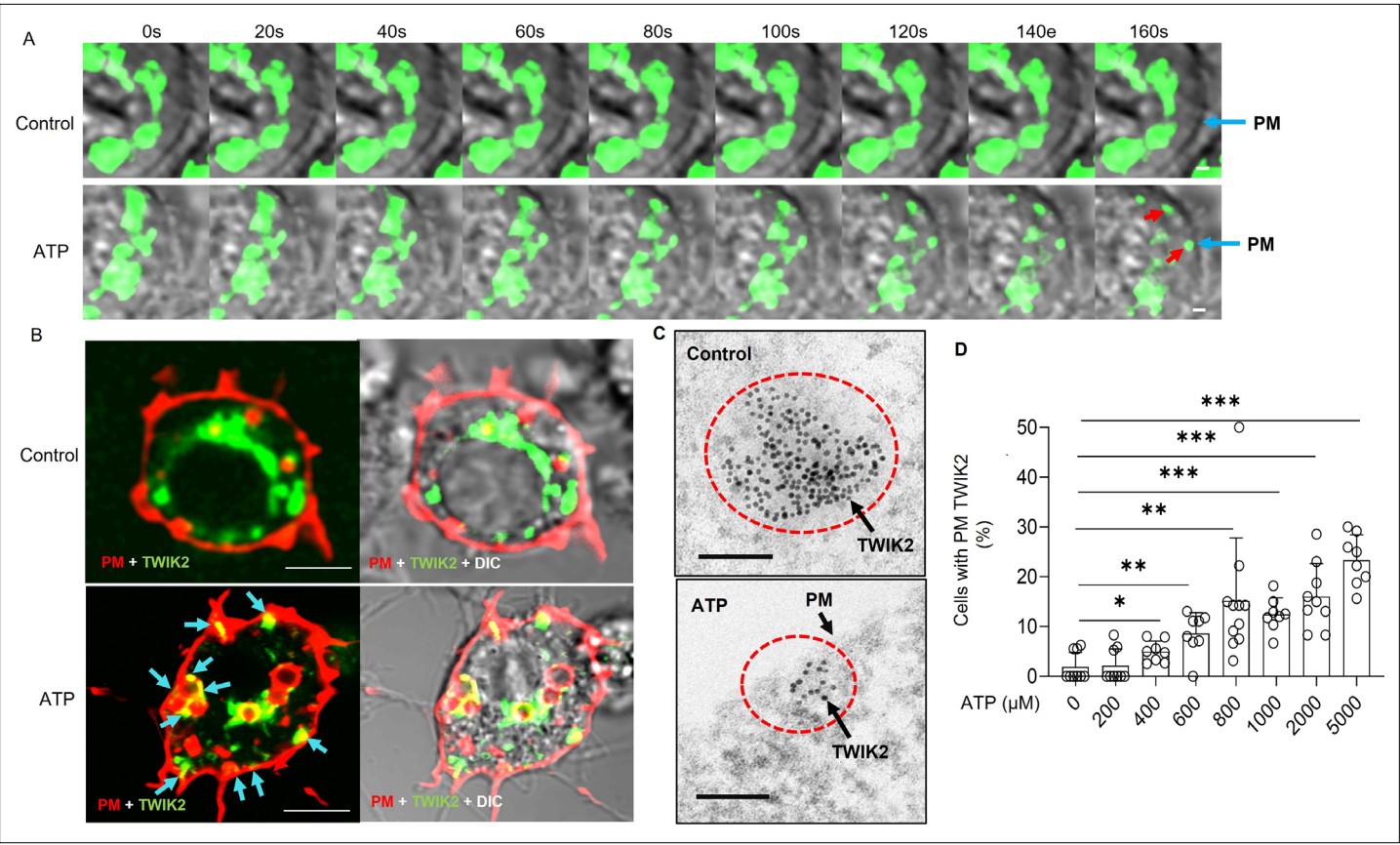

**Figure 1.** Intracellular TWIK2 plasmalemma translocation on ATP challenge. (**A**) Visualization of intracellular TWIK2 plasmalemma translocation post-ATP challenge. TWIK2-GFP plasmids were transfected into RAW 264.7 cells for 48 hr and cells were imaged with confocal microscope in the presence or absence of extracellular ATP (5 mM). Red arrows show translocated TWIK2 (green) on the plasma membrane (PM). Scale bar = 1 μm. (**B**) Confocal images of TWIK2 plasmalemma translocation on ATP challenge with the indicator of the plasma membrane (red) in mouse RAW 264.7 macrophage cell transfected with TWIK2-GFP plasmids (green). Cells expressing TWIK2-GFP were incubated with membrane dye NIR 750 (cell brite, #30077) for 20 m at 37°C followed by 2× wash with blank medium, stimulated with ATP or phosphate-buffered saline (PBS) (control) for 15 m; then cells were imaged using confocal microscope. Blue arrows showing the translocated TWIK2 (green) on the plasma membrane. Scale bar = 5 μm. (**C**) Confirmation of TWIK2 plasma membrane translocation using immunogold labeling electron microscopy before (upper panel) and after ATP (lower panel) (5 mM, 30 m) challenge in RAW 264.7 macrophages transfected with TWIK2 plasmids. TWIK2 (10 nm gold particles) was identified with anti-TWIK2 antibody (anti-aa71-120 of human TWIK2, Life Span Bioscience, LSBio #LS-C110195-100). Scale bar = 100 nm. Note vesicular structure outlined by immunogold marker in the upper panel and distribution of immunogold-labeled TWIK2 in the plasma membrane (PM) after ATP challenge in the lower panel. (**D**) ATP concentration-dependent TWIK2 plasma membrane translocation. TWIK2 plasma membrane (PM) translocation was analyzed using confocal images as shown in (**B**). *$p < 0.05$, **$p < 0.01$, ***$p < 0.001$. See also *Figure 1—figure supplement 1*.

The online version of this article includes the following video and figure supplement(s) for figure 1:

**Figure supplement 1.** Electron microscopy assessment of TWIK2 plasmalemma translocation.

**Figure 1—video 1.** Plasma membrane translocation of intracellular TWIK2 (control).
https://elifesciences.org/articles/83842/figures#fig1video1

**Figure 1—video 2.** Plasma membrane translocation of intracellular TWIK2 upon ATP challenge.
https://elifesciences.org/articles/83842/figures#fig1video2

of both IL-1β and IL-18, but not TNF-α, were significantly reduced in MDM treated with Vacuolin (*Figure 3C–I*). These results demonstrated the requisite role for plasmalemma–endosome fusion in mediating ATP-induced current and NLRP3 inflammasome activation.

Second, we determined the role of Ca²⁺ in mediating ATP-induced current and NLRP3 activation in MDMs. Cells pretreated with BAPTA-AM (1,2-Bis [2-aminophenoxy] ethane-*N,N,N′,N′*-tetraacetic acid tetrakis [acetoxymethyl ester]) significantly reduced ATP-induced current (*Figure 4A, B*). Cells subjected reduced extracellular Ca²⁺ or pretreated with BAPTA-AM showed significant reduction in NLRP3 inflammasome activation (*Figure 4C–L*) as reflected by the significantly reduced caspase 1

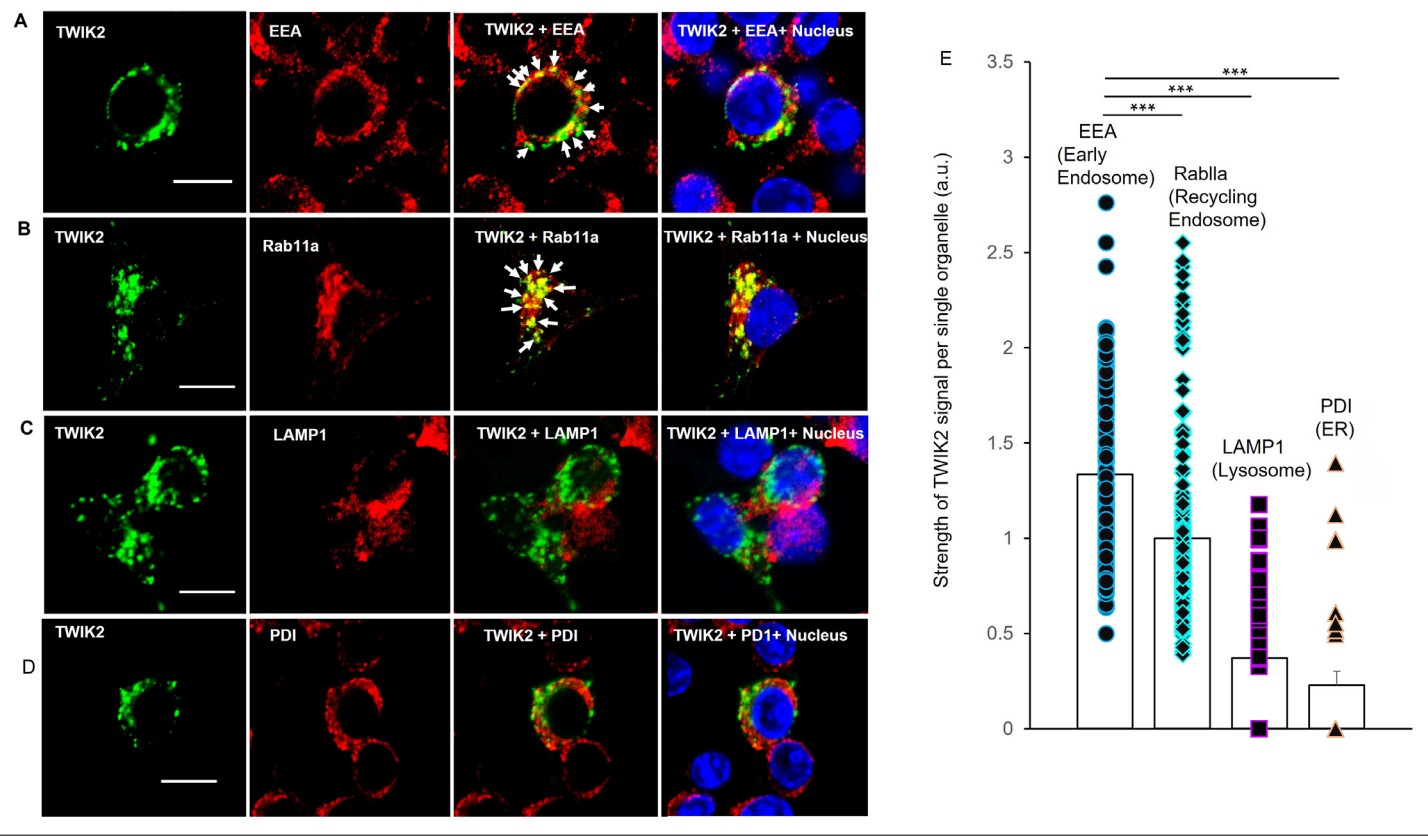

**Figure 2.** Endosomal localization of intracellular TWIK2 determined with fluorescent immunostaining of macrophages. TWIK2 intracellular localization was determined with fluorescent immunostaining with TWIK2 antibody along with other various antibodies against some specific vesicular proteins and imaged with confocal microscope. RAW 264.7 macrophages transfected with TWIK2 plasmids were fixed/permeabilized followed by immunostaining. TWIK2 (green) was identified with anti-TWIK2 antibody (LSBio #LS-C110195-100) in (**A–D**). Early endosomes (EE, red) were identified with antibody (clone1D4B) against EEA1 (C45B10 from Cell Signaling Technology) in (**A**). Recycling endosomes (RE, red) were identified with antibody against Rab11a (ab65200 from Abcam) in (**B**). Lysosomes (red) were identified with antibody against lysosomal membrane protein LAMP1 (D2D11 from Cell Signaling Technology) in (**C**). ER was identified with antibody against Protein Disulfide Isomerase (PDI; C81H6 from Cell Signaling Technology) in (**D**). Scale bar = 10 μm. (**E**) Quantification of co-localization of TWIK2 with cell organelles based on the confocal images as shown in (**A–D**). ***p < 0.001. The localization of TWIK2 with organelles (green) was seen in both the EE and RE (white arrows) but much less in lysosomes or ER.

activation and IL-1β maturation (*Figure 4C–L*) and also significantly reduced releases of IL-1β and IL-18 (*Figure 4I, J*), but not TNF-α (*Figure 4K*), induced with ATP (5 mM). These results demonstrate the requisite role for Ca²⁺ in ATP-induced current and NLRP3 inflammasome activation. Thus together they showed the key role of P2X7-mediated Ca²⁺ influx during ATP-dependent exocytosis linked to plasmalemma potassium efflux and NLRP3 inflammasome activation.

## Rab11a in RE is required for endosomal TWIK2 plasmalemmal translocation, sepsis-induced NLRP3 inflammasome activation, and lung inflammation

Since plasmalemmal translocation of endosomal TWIK2 involves endosomal fusion with the plasmalemma as described above, we focused on identifying the Ca²⁺-sensitive GTP-binding protein translocation and fusion machinery (Rab family; *Parkinson et al., 2014*; *Naslavsky and Caplan, 2018*; *Takahashi et al., 2012*), Synaptotagmin family (Syt; *Jahn and Fasshauer, 2012*), and vesicle-associated membrane proteins (Vamp, a.k.a. Synaptobrevin; *Logan et al., 2003*; *Meng et al., 2007*; *Quetglas et al., 2002*). Quantitative assessment of expression showed that mRNA expression of Rab11a was the maximal as compared to other genes involved in translocation and fusion of endosomes (*Figure 5—figure supplement 1*). Since Rab11a is generally thought to regulate the function of specific endosomal subpopulation, RE (*Yu et al., 2014*; *Welz et al., 2014*), we next examined the

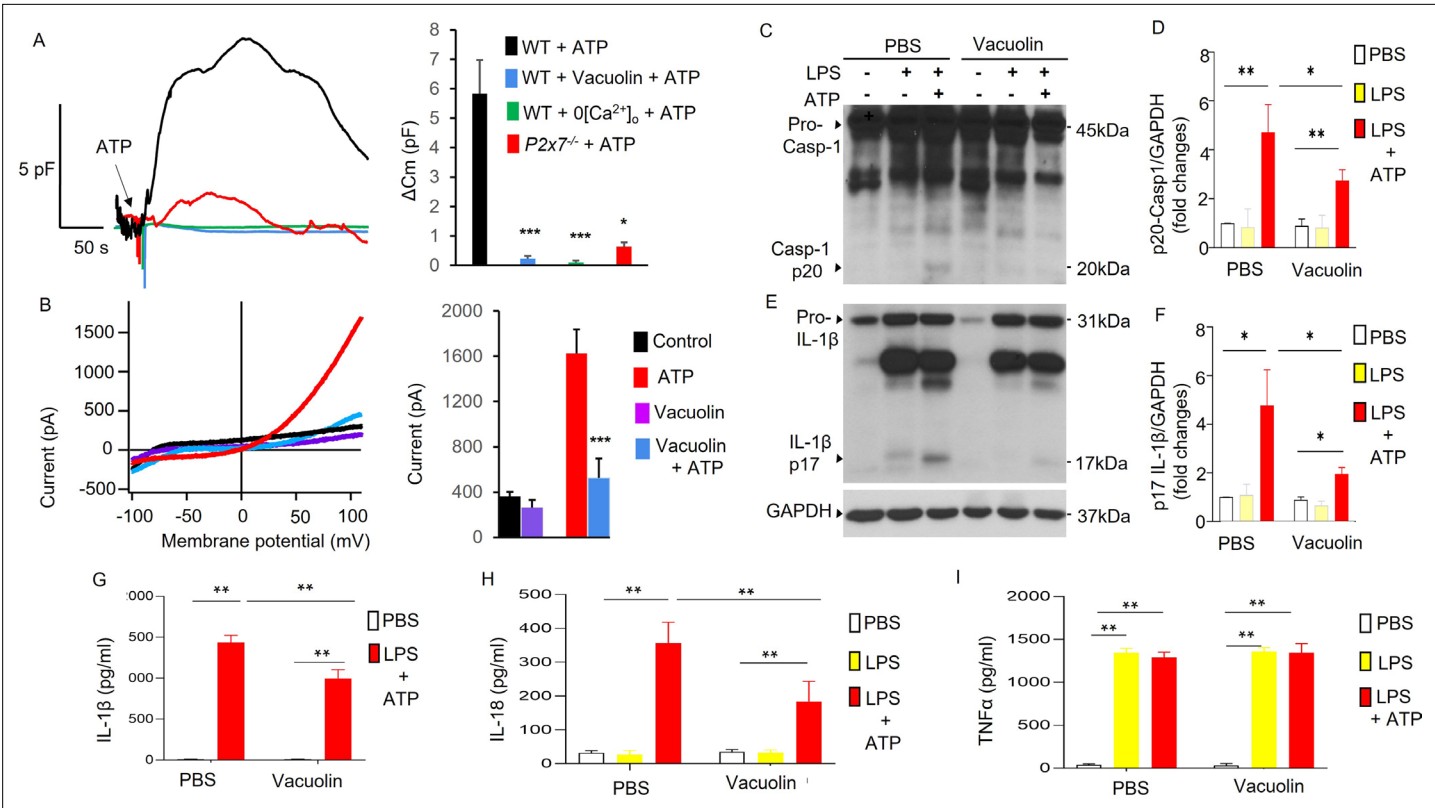

**Figure 3.** Association of P2X7-dependent ATP-induced exocytosis with plasmalemma potassium efflux and NLRP3 inflammasome activation. (**A, B**) Exocytic event is linked to plasmalemma potassium efflux. (**A**) Exocytosis was evaluated with measurement of whole-cell plasma membrane capacitance (Cm) reflecting the membrane surface changes. Left panel: Raw Cm traces recorded in monocyte-derived macrophages (MDMs) from either three WT or $P2x7^{-/-}$ mice under different conditions: 0 extracellular $Ca^{2+}$ ($0[Ca^{2+}]_o$) to confirm the involvement of P2X7 which mediated $Ca^{2+}$ influx; Vacuolin, an inhibitor of vesicle–plasma membrane fusion to confirm the involvement of membrane fusion events. 5 mM ATP was added as indicated. In the Vacuolin group, cells were treated with 10 µM Vacuolin for 2 hr before challenge with ATP. ATP treatment caused Cm increase indicating intracellular vesicle fusion with plasma membrane. Right panel: Summary of capacitance changes shown in left panel. ***$p < 0.001$ compared with WT + ATP group, ($n = 5$). Vesicle–membrane fusion inhibitor Vacuolin prevented ATP-induced exocytosis (Cm increase), indicating the increased Cm caused by ATP challenge is the result of fusion of intracellular vesicles with the plasma membrane. The fusion event is both P2X7 and extracellular $Ca^{2+}$ dependent. (**B**) Vesicle–plasmalemma fusion dependent of ATP-induced potassium efflux current. Whole-cell current was recorded with patch clamp in MDM from three mice with or without Vacuolin (10 µM). Currents were elicited with a ramp voltages running from −110 to +110 mV within 200 ms applied to cells with intervals of 1 s. Cells were held at 0 mV. Cells were bathed in solutions with $K^+$ as the major outward current and $Na^+$ and $Ca^{2+}$ as the major inward current. Left panel: Representative *I–V* plot of whole-cell current in MDM. Right panel: Summary of experiments displayed in left panel. ***$p < 0.001$ compared with ATP group ($n = 5$). Cells pretreated with the inhibitor of vesicular fusion protein Vacuolin showed significantly decreased current induced by ATP. (**C–I**) Inhibition of vesicle–plasmalemma fusion prevents NLRP3 inflammasome activation in macrophages (MDMs). Representative western blots from three independent experiments showing reduced caspase 1 activation (reduced Casp-1 p20; **C**) and IL-1β maturation (reduced IL-1β p17; **E**). MDMs from three mice pretreated with vesicle–plasmalemma fusion inhibitor Vacuolin (10 µM, 2 hr) were primed with lipopolysaccharide (LPS; 3 hr) and subsequently challenged with ATP (5 mM) for 30 min. Cell lysate was immunoblotted with indicated antibodies (anti-TWIK2 or anti-IL1β). (**D, F**) Quantification of results in (**C, E**). *$p < 0.05$, **$p < 0.01$, $n = 3$. Reduction in (Casp-1 p20) and IL-1β p17 was seen in cells treated with Vacuolin consistent with the results above. Reduction in release of IL-1β and IL-18 was also evident but TNF-α (tumor necrosis factor alpha) did not change in the presence of Vacuolin (shown in (**G, H**) and (**I**), respectively). *$p < 0.05$, **$p < 0.01$.

The online version of this article includes the following source data for figure 3:

**Source data 1.** P2X7-dependent ATP-induced exocytic event is linked to plasmalemma potassium efflux and NLRP3 inflammasome activation.

**Source data 2.** P2X7-dependent ATP-induced exocytic event is linked to plasmalemma potassium efflux and NLRP3 inflammasome activation.

role of Rab11a in mediating endosomal TWIK2 plasmalemma translocation. Here, we first determined the cellular distribution of Rab11a before and after ATP challenge using fluorescence immunostaining confocal microscopy. The images showed Rab11a plasmalemmal translocation after ATP challenge in MDMs (*Figure 5A, B*).

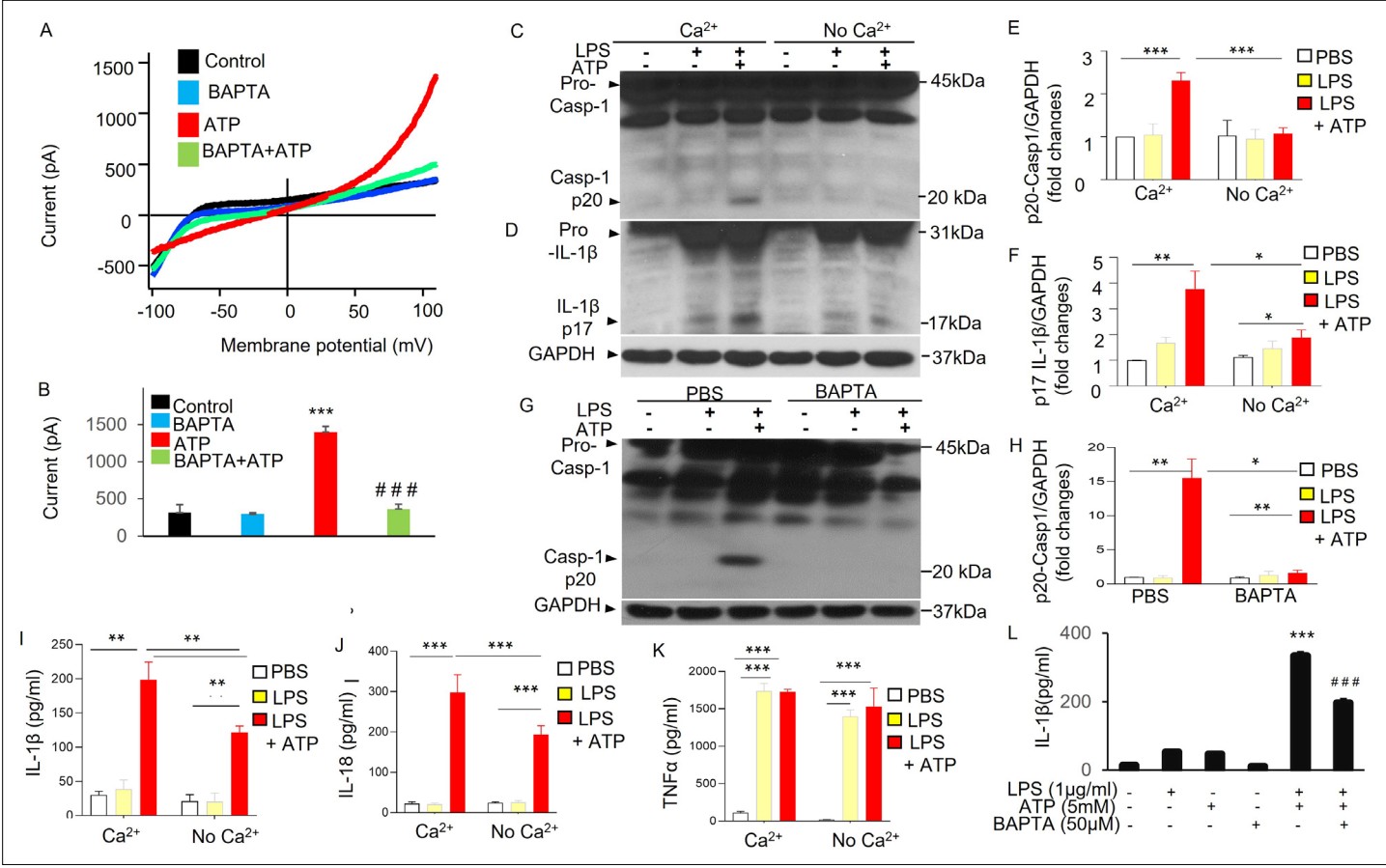

**Figure 4.** ATP-induced potassium current and NLRP3 inflammasome activation in $Ca^{2+}$-dependent manner. (**A, B**) $Ca^{2+}$ dependent of ATP-induced potassium efflux current. Whole-cell current was recorded with patch clamp in monocyte-derived macrophages (MDMs) from three mice with or without pretreatment of BAPTA-AM (1,2-Bis [2-aminophenoxy] ethane-*N,N,N′,N′*-tetraacetic acid tetrakis [acetoxymethyl ester], 10 μM) for 20 m. Currents were elicited with a ramp voltages running from −110 to +110 mV within 200 ms applied to cells with an interval of 1 s. Cells were held at 0 mV. Cells were bathed in solutions with $K^+$ as the major outward current and $Na^+$ and $Ca^{2+}$ as the major inward current. (**A**) Representative *I–V* plot of whole-cell current in MDMs. (**B**) Summary from experiments displayed in (**A**). ***p < 0.001 compared with control group, n = 7; ###p < 0.001 compared with ATP group, (n = 7). Cells pretreated with BAPTA-AM showed significantly decreased current induced by ATP (5 mM). (**C–F**) Extracellular $Ca^{2+}$-dependent NLRP3 inflammasome activation in macrophages. MDMs from three mice were primed with lipopolysaccharide (LPS) and subsequently challenged with ATP and cell lysates were immunoblotted with indicated antibodies (anti-Caspase 1 or anti-IL1β). (**C, D**) Representative western blotting results from three independent experiments showing reduced caspase 1 activation (reduced Casp-1 p20) and IL-1β maturation (reduced IL-1β p17) in the absence of extracellular $Ca^{2+}$. (**E, F**) Quantification of results shown in (**C, D**). *p < 0.05, **p < 0.01, ***p < 0.001, n = 3. The absence of extracellular $Ca^{2+}$ prevented ATP-induced NLRP3 inflammasome activation in MDMs. (**G, H**) Reduced caspase 1 activation in the presence of $Ca^{2+}$ chelator BAPTA-AM in MDMs. MDMs from three mice were primed with LPS (3 hr) and were then pretreated with or without BAPTA-AM (10 μM) for 30 m and subsequently challenged with ATP (5 mM) for 30 m and cell lysates were immunoblotted with anti-Caspase 1. (**G**) Representative western blotting results from three independent experiments showing reduced caspase 1 activation (reduced Casp-1 p20) on cell treatment with BAPTA-AM. (**H**) Quantification of results shown in (**G**). *p < 0.05, **p < 0.01, n = 3. Consistent with these results, the release of IL-1β and IL-18 but not TNF-α, was reduced in the absence of extracellular $Ca^{2+}$ or in the presence of $Ca^{2+}$ chelator BAPTA-AM as shown in (**I**), (**J**), (**K**), and (**L**), respectively. **p < 0.01, ***p < 0.001, n = 3.

The online version of this article includes the following source data for figure 4:

**Source data 1.** $Ca^{2+}$-dependent plasmalemma potassium efflux and NLRP3 inflammasome activation.

**Source data 2.** ATP-induced potassium current and NLRP3 inflammasome activation in $Ca^{2+}$-dependent manner.

**Source data 3.** ATP-induced potassium current and NLRP3 inflammasome activation in $Ca^{2+}$-dependent manner.

To assess the role of Rab11a along with P2X7 and $Ca^{2+}$ in ATP-induced TWIK2 plasmalemma insertion, we examined the cellular distribution of TWIK2 before and after ATP challenge using confocal microscopy in RAW 264.7 macrophages transfected with TWIK2-GFP plasmid. Cells in which Rab11a was depleted (with siRab11a) or cells in which P2x7 depleted (with siP2X7) or cells pretreated with BAPTA-AM, all showed significantly reduced TWIK2 (*Figure 5C–F*).

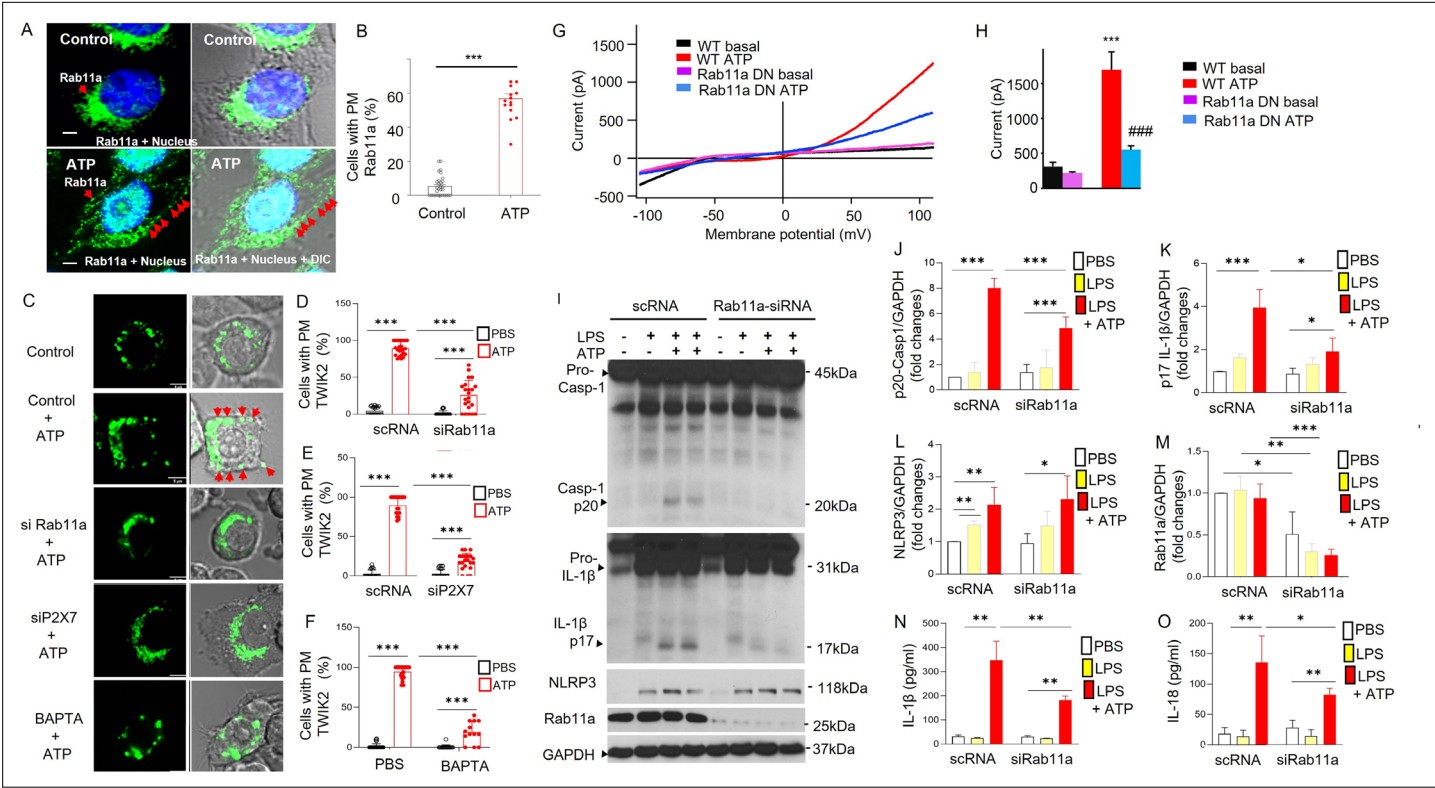

**Figure 5.** Rab11a mediates endosomal TWIK2 plasmalemma translocation and NLRP3 inflammasome activation on ATP challenge of macrophages. (**A**) Confocal images of Rab11a immunostaining in mouse monocyte-derived macrophages (MDMs) from three mice before and after ATP challenge. Rab11a (green) distribution was identified with fluorescent immunostaining with anti-Rab11a antibody (ab65200 from Abcam). Red arrows show translocated Rab11a after ATP challenge. Scale bar = 10 μm. We observed dispersed distribution and plasmalemma translocation of Rab11a after ATP challenge (bottom panel). (**B**) Summary of Rab11a plasmalemma translocation as shown in (**A**). ***$p < 0.001$ compared with control group. (**C–F**) Rab11a-, P2X7-, and $Ca^{2+}$-dependent TWIK2 plasma membrane translocation induced by ATP. (**C**) Confocal images of TWIK2 plasmalemma translocation on ATP challenge in mouse RAW 264.7 macrophage cells transfected with TWIK2-GFP plasmids (green) under different conditions as indicated. Cells pretreated with siRab11a (or scRNA as control) or siP2X7 (or scRNA as control) for 48 hr or pretreated with BAPTA-AM (or phosphate-buffered saline [PBS] as control) for 20 m were stimulated with ATP or PBS (control) for 15 m; then the cells were imaged using a confocal microscope. Red arrows showing the translocated TWIK2 after ATP challenge. Scale bar = 5 μm. (**D–F**) Quantification of the TWIK2 plasmalemma translocation under different conditions (pretreated with siRab11a (**D**), pretreated with siP2X7 (**E**), and pretreated with BAPTA-AM (**F**)) based on the confocal images as shown in (**E**). ***$p < 0.001$. Depletion of Rab11a or P2x7 or blocking intracellular $Ca^{2+}$ increase (by BAPTA-AM) significantly reduced TWIK2 plasmalemma translocation after ATP challenge. (**G, H**) Reduced ATP-induced $K^+$ outward current in RAW 264 macrophages treated with dominant-negative Ra11a (Rab11a DN) for 48 hr. Whole-cell current was recorded with patch clamp as described in *Figure 3B*. (**G**) Representative *I–V* plot of whole-cell current. (**H**) Summary from experiments displayed in (**G**). ***$p < 0.001$ compared with WT basal, $n = 5$. ###$p < 0.001$ compared with WT ATP group, $n = 5$. Cells pretreated with Rab11a DN showed significantly decreased current induced by ATP. (**I–O**) Rab11a-dependent NLRP3 inflammasome activation induced by ATP in macrophages. These experiments were carried out in MDMs treated with siRNA targeting mouse Rab11a (siRab11a). (**I**) Representative western blot results from three independent experiments with MDMs from three mice showing reduced caspase 1 activation (reduced Casp-1 p20), IL-1β maturation (reduced IL-1β p17), and depletion of Rab11a in cells treated with siRab11a in MDMs whereas NLRP3 expression was not affected by siRab11a. MDMs pretreated with siRab11a for 48 hr were primed with lipopolysaccharide (LPS; 3 hr) and subsequently challenged with ATP (5 mM) for 30 m. Cell lysates were immunoblotted with indicated antibodies (anti-TWIK2 or anti-IL1β or anti-Rab11a or anti NLRP3). (**J–M**). Quantification of results shown in (**I**). *$p < 0.05$, **$p < 0.01$, ***$p < 0.001$, $n = 3$. The reductions in Casp-1 p20, IL-1β p17, Rab11a, but not NLRP3 expression were seen in cells treated with siRab11a. Reduced IL-1β (**N**) and IL-18 (**O**) release in cells treated with siRab11a. MDMs pretreated with siRab11a for 48 hr were primed with LPS (3 hr) and subsequently challenged with ATP (5 mM) for 30 m. Release in IL-1β and IL-18 in the supernatant was measured with ELISA. *$p < 0.05$, **$p < 0.01$, $n = 3$. See also *Figure 5—figure supplement 1*.

The online version of this article includes the following source data and figure supplement(s) for figure 5:

**Source data 1.** Rab11a mediates endosomal TWIK2 plasmalemma translocation and NLRP3 inflammasome activation on ATP challenge in macrophages.

**Figure supplement 1.** Relative mRNA expression of vesicle fusion proteins in monocyte-derived macrophages (MDMs).

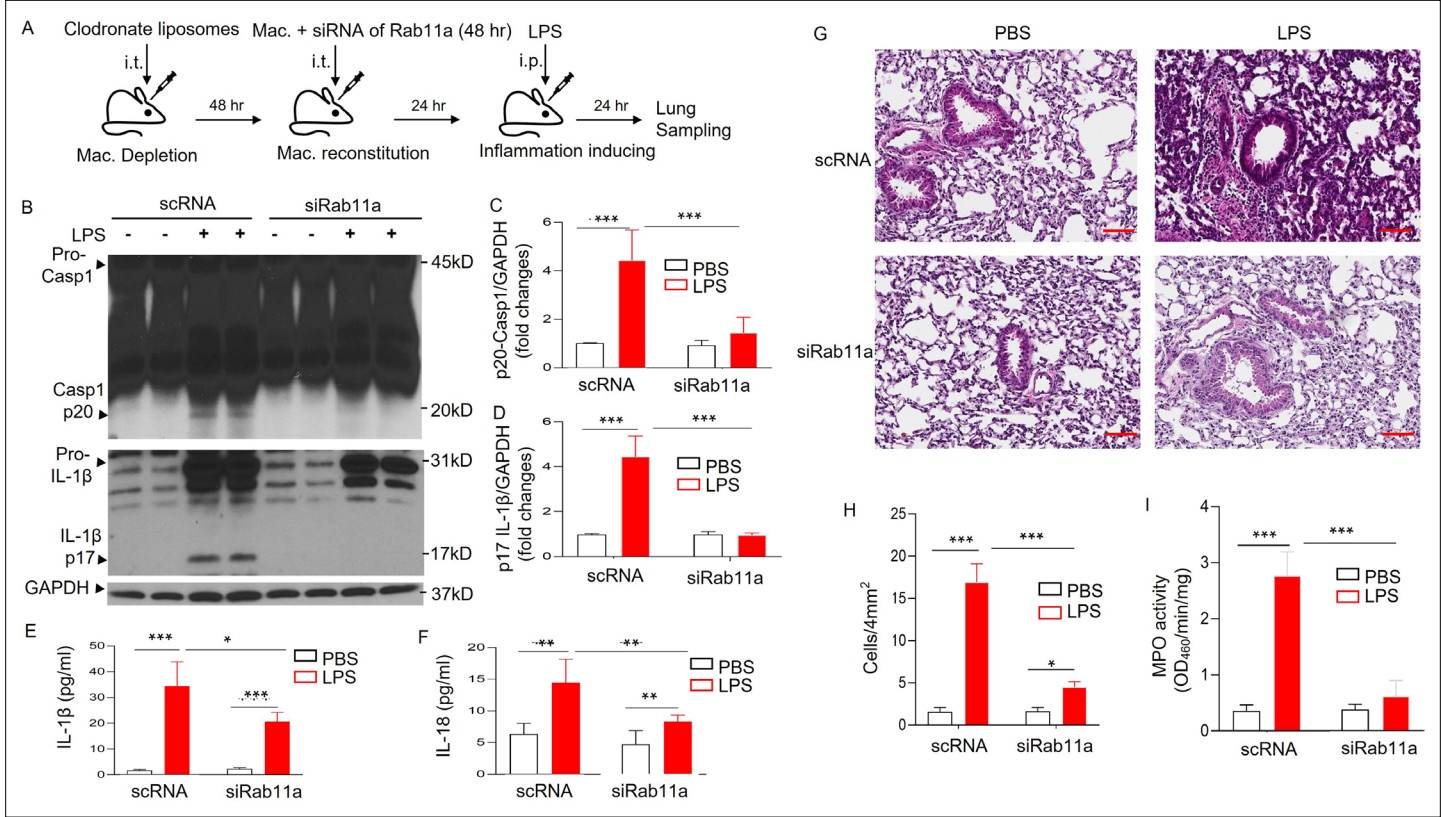

**Figure 6.** Rab11a deficiency in macrophages prevents sepsis-induced NLRP3 inflammasome activation and inflammatory lung Injury in mice. (**A**) Schematic illustration of the experiments. Lung macrophages (Mac) were depleted with clodronate liposomes (for 48 hr) and then reconstituted intratracheally with monocyte-derived macrophages (MDMs) treated with either siRNA of Rab11a or scRNA control as illustrated each group (5 mice per group) were injected with lipopolysaccharide (LPS; intra-peritoneal injection, i.p.) after 24 hr of macrophage reconstitution. Lungs were harvested for evaluation of NLRP3 inflammasome activation and lung inflammation. NLRP3 inflammasome activation was evaluated by both measuring the density of Casp-1 p20 and IL-1β p17 based on immunoblotting (**B**) and quantified in (**C** and **D**) (***p < 0.001, n = 5), and measuring the concentrations of IL-1β and IL-18 shown in (**E**) (IL-1β) and (**F**) (IL-18), *p < 0.05, **p < 0.01, ***p < 0.001, n = 5. (**G**) Representative H&E images of lung sections from three independent experiments (scale bars: 200 μm). Lung injury shown in (**G**) was evaluated by quantification of inflammatory cells in alveoli (per 4 mm² using the *Fiji* image analysis software) shown in (**H**) (*p < 0.05, ***p < 0.001, n = 5). Lung neutrophil infiltration was evaluated by MPO measurements of lung tissue shown in (**I**) (***p < 0.001, n = 5). See also *Figure 6—figure supplement 1*.

The online version of this article includes the following source data and figure supplement(s) for figure 6:

**Source data 1.** Rab11a deficiency in macrophages prevents sepsis-induced NLRP3 inflammasome activation and inflammatory lung injury in mice.

**Figure supplement 1.** Endosomal TWIK2 plasmalemma translocation and resultant NLRP3 inflammasome activation.

We next tested the role of Rab11a in ATP-induced current and NLRP3 inflammasome activation. Here, we first measured the ATP-induced K⁺ current in RAW 264 macrophages expressing dominant-negative Rab11a construct (Rab11a S25N). Cells treated with Rab11a S25N showed significantly reduced ATP-induced K⁺ current (*Figure 5G, H*), indicating the requisite role for Rab11a in inducing K⁺ current. Depletion of Rab11a with siRNA (*Figure 5I, M*) in MDMs followed by assessment of NLRP3 inflammasome activation by ATP (5 mM) showed reductions in caspase 1 activation and IL-1β maturation (*Figure 5I–K*) and in the release of IL-1β and IL-18 (*Figure 5N, O*), whereas NLRP3 expression was not affected (*Figure 5I, L*).

To identify the in vivo role of macrophage-expressed Rab11a in regulating NLRP3 inflammasome activation and inflammation, we first depleted endogenous mouse lung macrophages with liposomal clodronate (*Weisser et al., 2012*), and carried out adoptive transfer (via i.t. route) of MDMs with and without siRNA-mediated Rab11a depletion (*Figure 6A*). We then induced endotoxemia in recipient mice. Mice transplanted with Rab11a-depleted macrophages showed significantly reduced NLRP3 inflammasome activation (*Figure 6B–F*) reflected by reduced caspase 1 activation and IL-1β maturation (*Figure 6B–D*) as well as the release of IL-1β and IL-18 (*Figure 6E, F*). Severity of inflammatory

lung injury in mice with Rab11a-depleted macrophages was significantly reduced as assessed by neutrophil and macrophage infiltration in lungs (*Figure 6G, H*) and quantification of myeloperoxidase (MPO) activity in lungs (*Figure 6I*).

## Discussion

The inflammasomes as the innate immune signaling receptors monitor the extracellular space and subcellular compartments for signs of infection, damage, and other cellular stressors (*Latz et al., 2013*). NLRP3 inflammasome is a protein complex consisting of the inflammasome sensor molecule NLRP3, the adaptor protein ASC, and caspase 1 (*Welz et al., 2014*; *Weisser et al., 2012*). NLRP3 formation is triggered by a range of substances generated during infection, tissue damage, and metabolic imbalances (*Latz et al., 2013*; *Schroder and Tschopp, 2010*); however details of NLRP3 activation by events at the plasma membrane remain unclear. NLRP3 activation is comprised of an initial priming phase involving NF-κB-dependent transcription of NLRP3 and pro-interleukin-1β release initiated by pro-inflammatory cytokines or stimulation of Toll-like receptor (TLR) by agonists such as lipopolysaccharide (LPS) (*Burns et al., 2003*). The second phase of NLRP3 activation is initiated by Pathogen-Associated Molecular Patterns (PAMPs) or Danger-Associated Molecular Patterns (DAMPs) such as ATP, which ligates the purinergic receptor P2X7 (*Surprenant et al., 1996*). Based on studies on the structure and assembly mechanisms of NLRP3 complex, an endogenous, stimulus-responsive form of full-length mouse NLRP3 is a 12- to 16-mer double-ring cage held together by leucine-rich-repeat (LRR)–LRR interactions with the pyrin domains shielded within the assembly to avoid premature activation (*Andreeva et al., 2021*). NLRP3 (consisting of double-ring cages of NLRP3) is predominantly membrane associated, such as in ER, mitochondria, Golgi apparatus (*Zhou et al., 2011*; *Subramanian et al., 2013*; *Wang et al., 2020*), and trans-Golgi network dispersed vesicles (dTGNvs), an early event observed in response to NLRP3-activating stimuli (*Andreeva et al., 2021*). Double-ring caged NLRP3 is recruited to the dispersed TGN (dTGN) through ionic bonding between its conserved polybasic region and negatively charged phosphatidylinositol-4-phosphate (PtdIns4P) on the dTGN. dTGNvs serves as a scaffold for NLRP3 aggregation into multiple puncta, leading to polymerization of the adaptor protein ASC, and thereby activates the downstream signaling cascade (*Chen and Chen, 2018*). NLRP3 oligomer on the membrane is poised to sense diverse signals and induce inflammasome activation (*Andreeva et al., 2021*). NLRP3 inflammasome is also assembled and activated at the centrosome (*Wu et al., 2022*; *Li et al., 2017*; *Yang et al., 2020*; *Magupalli et al., 2020*), the microtubule organizing center in mammalian cells, accounting for the singularity, size, and perinuclear location of activated inflammasomes (*Wu et al., 2022*). However, despite understanding these pathway, little is known about the triggers at the plasma membrane initiating the downstream assembly and activation of NLRP3 complex.

Although it is known that a key upstream trigger of NLRP3 assembly and activation is potassium efflux (*Pétrilli et al., 2007*; *Franchi et al., 2007*; *Muñoz-Planillo et al., 2013*) via the potassium channel TWIK2 (*Di et al., 2018*) which creates regional pockets of low potassium, and thus facilitates NLRP3 assembly, the role of potassium efflux in the activation of the NLRP3 complex is unknown. It was reported that cellular K$^+$ efflux stabilized structural change in the inactive NLRP3, promoting an open conformation as a step preceding activation (*Tapia-Abellán et al., 2021*). The conformational change appeared to facilitate the assembly of NLRP3 into a seed structure for ASC oligomerization, a key step for NLRP3 inflammasome activation (*Tapia-Abellán et al., 2021*). In the present study, we examined how cells are able to control and modulate activation of the NLRP3 inflammasome despite TWIK2 being a continuously active background K$^+$ channel (*Enyedi and Czirják, 2010*).

Electrophysiological characterization and understanding of the functional significance of TWIK channel family (TWIK1, TWIK2, and TWIK7) is impeded by the low or absent functional expression in heterologous expression systems (*Enyedi and Czirják, 2010*). TWIK1 was reported to be mainly located in intracellular compartments (such as pericentriolar RE), and its transfer to the plasma membrane is tightly regulated (*Enyedi and Czirják, 2010*). A study also showed that the TWIK2 generated background K$^+$ currents in endolysosomes was crucial in regulating the number and size of lysosomes in MDCK cells (*Bobak et al., 2017*). TWIK2 contains sequence signals responsible for the expression of TWIK2 in the Lamp1-positive lysosomal compartment (*Bobak et al., 2017*), and sequential inactivation of these trafficking motifs prevented the targeting of TWIK2 to lysosomes, thus enabling plasmalemmal relocation of the functional channel (*Bobak et al., 2017*). Here, we

determined the mechanism of TWIK2 expression at the plasmalemma to address how the channel is functionalized in macrophages in the face of high intracellular $K^+$ concentration. We showed that TWIK2 in macrophages was primarily distributed in endosomes, but importantly it translocated on demand by ATP within 2 min to the plasmalemma by Rab11a-dependent mechanism. We showed that TWIK2 was primarily located in the endosomal compartment at rest, thus preventing TWIK2-mediated $K^+$ efflux into the extracellular space to avoid unchecked NLRP3 activation. However, upon ligation of the purinergic P2X7 receptor by extracellular ATP, $Ca^{2+}$ influx via P2X7 activated the $Ca^{2+}$-sensitive endosomal GTPase Rab11a to induce endosomal TWIK2 translocation to the plasma membrane. $K^+$ efflux via plasma membrane translocated TWIK2 caused local $K^+$ concentration ($[K^+]_{in}$) to decrease leading to NLRP3 inflammasome activation and phenotype transition of macrophages (*Figure 6— figure supplement 1*). In the model, the endosomes served as reservoirs for the ion channel and thus enabling their transport to plasmalemma upon stimulation.

There is precedence for this model. An increase in plasmalemma surface expression of G-protein-activated inwardly rectifying $K^+$ (GIRK) channels from RE functioned to modulate neuronal activity (*Chung et al., 2009b*; *Chung et al., 2009a*; *Grant and Donaldson, 2009*). RE also served as intracellular storage compartments for the cardiac pacemaker channels – hyperpolarization-activated cyclic nucleotide-gated (HCN) ion channels HCN2 and HCN4 for rapid adaptation of their surface expression in response to extracellular stimuli (*Hardel et al., 2008*).

Endosomal membrane trafficking requires the coordination of multiple signaling events to control cargo sorting and processing and endosome maturation (*Li et al., 2013*). Several key regulators have been identified in endosome trafficking, such as small GTPases (which initiate signaling cascades to regulate the direction and specificity of endosomal trafficking), Ca (*Broz and Dixit, 2016*), and phosphoinositides (*Li et al., 2013*). Here, we found that the GTPase Rab11a showed the highest expression in macrophages. Rab11a is thought to regulate the function of a special endosomal subpopulation, the RE (*Yu et al., 2014*; *Welz et al., 2014*). This heterogeneous tubular-vesicular compartment engaged in membrane trafficking, connects the endo- and exocytotic pathways (*Yu et al., 2014*; *Welz et al., 2014*). Although Rab11 is prominent in RE, other studies have addressed its role in intracellular domains, trans-Golgi network (TGN) and post-Golgi secretory vesicles (*Grant and Donaldson, 2009*; *Yu et al., 2014*; *Welz et al., 2014*). Rab11a plays a key role in mouse embryonic development through regulating the secretion of soluble matrix metalloproteinases required for cell migration, embryonic implantation, tissue morphogenesis, and innate immune responses (*Yu et al., 2014*). Rab11a-null embryos formed normal blastocysts but died at peri-implantation stages (*Yu et al., 2014*). We have previously described the role for Rab11a in regulating efferocytosis via the modulation of disintegrin and metalloproteinase (ADAM)17-mediated CD36 cell surface expression as a promising strategy for activating the resolution of inflammatory lung injury (*Jiang et al., 2017*). The present results show an obligatory role of Rab11a in mediating cycling endosomal TWIK2 plasmalemma translocation. Importantly as a test of Rab11a relevance, adoptive transfer of Rab11a-deleted macrophages into mouse lungs after alveolar macrophage depletion prevented NLRP3 inflammasome activation and inflammatory lung injury. Thus, Rab11a in macrophages has a fundamental check-point role in TWIK2 plasmalemmal translocation and regulating NLRP3 inflammasome activation and endotoxemia-induced inflammatory lung injury.

## Materials and methods
### Mice, cell cultures, and reagents
C57 black 6 (C57BL/6) mice were obtained from Charles River Laboratory. *Twik2*$^{-/-}$ mice was a generous gift from Dr. Lavannya M. Pandit (Baylor College of Medicine) (*Lloyd et al., 2011*; *Pandit et al., 2014*). *P2x7*$^{-/-}$ (B6.129P2-P2*rx*7tm1Gab/J) mice were purchased from Jackson Laboratory (stock number: 005576). All mice were housed in the University of Illinois Animal Care Facility in accordance with institutional and NIH guidelines. Veterinary care and animal experiments were approved by the University of Illinois Animal Care & Use Committee (ACC protocol number: 21-032). For LPS-induced injury, mice received a single intraperitoneal dose (20 mg/kg) of LPS (*Escherichia coli* 0111:B4, L2630, Sigma). Mouse bone marrow MDMs were induced and cultured as described (*Zhang et al., 2008*). The mouse RAW 264.7 macrophage cell line was obtained from ATCC (TIB-71). RAW 264.7 is a macrophage cell line that was established from a tumor in a male mouse. The identity of this cell line was authenticated

with the method of STR profiling by ATCC and was authenticated by ATCC, and was tested negative for mycoplasma contamination. These cells were cultured and propagated as instructed by the manufacturer's protocol. Caspase 1 antibody (p20, AG-20B-0042-C100) was purchased from AdipoGen Life Sciences. IL-1β antibody (AF-401-NA) was purchased from R&D Systems. TWIK2-EGFP plasmid (pLV[Exp]-Puro-CMV>mKcnk6[NM_001033525.3]/3xGS/EGFP) was designed by and purchased from VectorBbuilder (VB200618-1166ypg). TWIK2 antibody (LS-C110195-100) was purchased from Life Span Bioscience. Antibody against Rab11a was purchased from Abcam (ab65200). Antibody against EEA1 (C45B10), LAMP1 (D2D11), and PDI (C81H6) were purchased from Cell Signaling Technology. LPS (*E. coli* 0111:B4, Ultrapure, tlrl-3pelps, used to treat cells) were obtained from Invitrogen. ATP-Na$^+$ (A2383), Vacuolin-1(673000), and other chemicals were purchased from Sigma.

## Dynamic observations of intracellular TWIK2 plasmalemma translocation in macrophages

Time-lapse video recording with confocal microscope was used to follow intracellular TWIK2 plasmalemma translocation in macrophages challenged with ATP. TWIK2-GFP plasmid were transfected into RAW 264.7 cells for 48 hr and cells were imaged with confocal microscope in the presence or absence of extracellular ATP. Video recording was initiated once cells were exposed to ATP (5 mM) using a Zeiss LSM 710 confocal microscope using 488 nm lasers and a ×63 objective lens (NA 1.3) and an emission bandwidth of 500 nm. Dynamic analysis of intracellular TWIK2 plasmalemma translocation was performed using Fiji software.

## TWIK2 immunostaining in macrophages

RAW 264.7 macrophages transfected with TWIK2 plasmids were plated at a density of 300,000 cells on 25 mm coverslips. Following a 24-hr incubation, cells were treated with different concentrations of ATP for 30 min, followed by five washes with phosphate-buffered saline (PBS), and fixation for 20 min in 3% paraformaldehyde–PBS. Cells were blocked and permeabilized in 0.25% fish skin gelatin (Sigma), 0.01% saponin (Calbiochem, San Diego, CA) in PBS for 30 min. Cells were stained with TWIK2 antibody for 1 hr, coverslips were washed, then incubated with goat antirabbit-AlexaFluor488 (Molecular Probes) for 1 hr, washed again and mounted in 4% *n*-propyl gallate, 25 mM Tris at pH 8.5 and 75% glycerol. Related localizations of TWIK2 were imaged with a Zeiss LSM 710 confocal microscope using 488 and 561 nm lasers and a ×63 objective lens (NA 1.3) and an emission bandwidth of 500–535 nm. Images were acquired with LCS software and images were processed with Fiji software.

## TWIK2 localization by immunoelectron microscopy

RAW 264.7 macrophages transfected with TWIK2 plasmids were collected and fixed by 4% paraformaldehyde and 0.15% glutaraldehyde in 0.1 M PB buffer for 1 hr. Cells were subsequently washed with 0.1 M PB buffer, dehydrated with ethanol and embedded with L.R. White resin (Electron Microscopy Science, Hatfield, PA) in a vacuum oven at 45°C for 48 hr. Sections (100 nm) were incubated with anti-TWIK2 primary antibody (LSBio, #LS-C110195-100) for 3 hr and further incubated with 10 nm gold-conjugated secondary antibody, goat anti-rabbit IgG (H+L; Ted Pella Inc, Redding, CA) for 1 hr. Cells were further stained with uranyl acetate and lead citrate and examined on a FEI Tecnai F30 at 300 kV.

## Whole-cell recordings

Electrophysiological recordings were obtained using a voltage-clamp technique. All experiments were conducted at room temperature (22–24°C) using an EPC-10 patch clamp amplifier (HEKA Electronik GmbH, Lambrecht, Germany) and using the Pulse V 8.8 acquisition program (HEKA Electronik GmbH, Lambrecht, Germany). Whole-cell currents were elicited by using a ramp protocol with test pulse range from −110 to +110 mV (200 ms in duration). The holding potential was 0 mV. The pipette solution contained (in mM): 120 K-glutamic acid, 2 Ca-acetate hydrate, 2 MgSO$_4$, 33 KOH, 11 EGTA [ethylene glycol-bis(β-aminoethyl ether)-N,N,N′,N′-tetraacetic acid], and 10 HEPES (N-2-hydroxyethylpiperazine-N-2-ethane sulfonic acid), pH 7.2. The bath solution contained (in mM): 140 Na-glutamic acid, 2 Ca-acetate hydrate, 1 MgSO$_4$, 10 HEPES, pH 7.4. Whole-cell capacitance was recorded as described (*Di et al., 2002*). Whole-cell currents and capacitance were analyzed using IGOR software (WaveMetrics, Lake Oswego, OR).

## Quantitative RT-PCR (reverse transcription-polymerase chain reaction) for fusion protein expression

Total RNA of cultured MDMs was extracted using the RNeasy Micro Kit (QIAGEN) according to the manufacturer's instructions. RNA isolated from MDMs was converted to cDNA using the High-Capacity cDNA Reverse Transcription Kit (Applied Biosystems). Real-time PCR was performed using SYBR Green Master Mix on ViiAZ (Applied Biosystems) according to the manufacturer's protocols. The following primers were used for PCR: Rab6a: forward, 5′-GATACTGCGGGTCAGGAACG-3′, and reverse, 5′-GCAGCAGAGTCACGGATGTAA-3′; Rab6b: forward, 5′-AACCCGCTGCGAAAATTCAA G-3′, and reverse, 5′-CGGTCTTCCAAGTACATGGTTT-3′; Rab11a: forward, 5′- AGGAGCGGTACA GGGCTATAA-3′, and reverse, 5′- ATGTGAGATGCTTAGCAATGTCA-3′; Rab11b: forward, 5′-GCTG CGGGATCATGCAGATAG-3′, and reverse, 5′- CACGGTCAGCGATTTGCTTC-3′; Rab27a: forward, 5′-GGCAGGAGAGGTTTCGTAGC-3′, and reverse, 5′-GCTCATTTGTCAGGTCGAACAG-3′; Rab27b: forward, 5′-TGGCTGAAAAATATGGCATACCA-3′, and reverse, 5′-CCAGAAGCGTTTCCACTGAC T-3′; Synaptotagmin VII-1 (Syt7-1): forward, 5′-TTGGCTACAACTTCCAAGAGTCC-3′, and reverse, 5′-CGGGTTTAGATTCTTCCGCTTC-3′; Syt7-2: forward, 5′-CAGACGCCACACGATGAGTC-3′, and reverse, 5′-CTGGTAAGGGAGTTGACGAGG-3′; Vapm2: forward, 5′-GCTGGATGACCGTGCAGAT-3′, and reverse, 5′-GATGGCGCAGATCACTCCC-3′; V amp3: forward, 5′-CAGGTGCCTCGCAGTTTGAA -3′, and reverse, 5′-CCTATCGCCCACATCTTGCAG-3′. The data were analyzed using the comparative cycle-threshold (CT) method, where the amount of target is normalized to an endogenous reference gene, GAPDH.

## Silencing Rab11a constructs

Dominant-negative Rab11a (Rab11a S25N) was a gift from Dr. Guochang Hu (The University of Illinois at Chicago) (*Jiang et al., 2017*). The siRNA targeting mouse Rab11a (L-040863-01-0005) and a siRNA-negative control were obtained from Horizon Discovery Ltd. Transient transfections of these dominant-negative Rab11a and siRab11a into mouse MΦs (RAW 264.7 cell line) were performed with Amaxa mouse macrophage nucleofector kit (VPA-1009, Lonza) and DharmaFECT 4 Transfection Reagent (T-2004-01, Dharmacon) according to the manufacturer's protocol. To evaluate the efficiency of Rab11a silencing, Rab11a expression was examined with western blot and inflammasome activation was examined by measuring p20 intensity via western blot and IL-1β release through ELISA as mentioned above in cells 2–3 days after transfection.

## NLRP3 inflammasome activation analysis

Prior to experimental treatments, macrophages incubated with 1 µg/ml LPS (LPS-EB Ultrapure, tlrl-3pelps, InvivoGen) as priming signal to induce NF-κB-dependent upregulation of pro-IL-1β and NLRP3 expression (*Katsnelson et al., 2015*). The cells were primed with LPS for 3 hr at 37°C and then priming medium was replaced with normal culture medium. To evaluate the NLRP3 inflammasome activation, macrophages were stimulated with 5 mM ATP for 30 min at 37°C and then IL-1β and IL-18 release in the medium or bath solutions were measured by ELISA and caspase 1 activation and IL-1β maturation were evaluated by western blot using p20 antibody of caspase 1 or IL-1β antibody. Briefly, cell-free supernatants were collected and then assayed for murine IL-1β (MLB00C, R&D Systems), IL-18 (7625, Medical Biological Lab), and TNF-α (SMTA00B, R&D Systems) by ELISA kit according to the manufacturer's protocol and the adherent macrophages were collected to generate whole-cell lysate. The cytokine concentration of the properly diluted or undiluted samples in 96-well plates was measured at 450 nm wavelength of absorbance, and calculated by GraphPad Prism linear regression analysis. Cell lysate samples and lung protein samples from mice were subjected to sodium dodecyl sulfate–polyacrylamide gel electrophoresis and transferred to membrane for western blot analysis using various primary antibodies (Caspase 1-p20 and IL-1β).

## MPO assay

The lungs were homogenized in 1 ml of PBS with 0.5% hexadecyltrimethylammonium bromide. The homogenates were sonicated, centrifuged at 40,000 × *g* for 20 min, and run through two freeze–thaw cycles. The samples were homogenized and centrifuged a second time. The supernatant was then collected and mixed 1/30 (vol/vol) with assay buffer (0.2 mg/ml o-dianisidine hydrochloride and

0.0005% $H_2O_2$). The change in absorbance was measured at 460 nm for 3 min, and MPO activity was calculated as the change in absorbance over time.

## Alveolar macrophage depletion and reconstitution

Commercially available clodronate liposomes (Clodrosome) were administered directly into lungs of 10-week-old mice using a minimally invasive endotracheal instillation method. The mice were anesthetized by ketamine and xylazine (45 and 8 mg/kg, respectively) and were suspended on a flat board and placed in a semi-recumbent position with the ventral surface and rostrum facing upwards. Using curved blade Kelly forceps, the tongue is gently and partially retracted rostrally and 50 μl of clodronate liposomes is placed in the back of the oral cavity, which is then aspirated by the animal. Control liposomes (50 μl) alone were similarly administered in the control group. After 2 days of clodronate treatment, mice were reconstituted by i.t. instillation in a similar manner with differentiated MDMs (these MDMs have been treated with siRab11a for 48 hr) at dose of $2 \times 10^6$ in a 50 μl volume per mice. The mice were injected with i.p. LPS (20 mg/kg) after 24 hr of macrophage reconstitution. The lungs were flushed and harvested after 24 hr of LPS challenge.

## Statistical analysis

Statistical comparisons were made using two-tailed Student's *t*-test for comparisons of two groups or one-way ANOVA followed by the Tukey's post hoc pairwise multiple comparisons when appropriate with Prism 6 (GraphPad). Experimental values were reported as the means ± standard error of the mean. Significance between groups was determined using the *t*-test (two tails) and asterisks indicate a statistically significant difference with the number of experiments indicated in parentheses.

## Acknowledgements

The work was supported by NIH grants P01-HL60678, P01-HL077806, T32-HL007829, R01-HL118068, and R01-HL90152.

## Additional information

### Funding

| Funder | Grant reference number | Author |
|---|---|---|
| National Institutes of Health | P01-HL60678 | Asrar B Malik |
| National Institutes of Health | P01-HL077806 | Asrar B Malik |
| National Institutes of Health | R01-HL118068 | Asrar B Malik |
| National Institutes of Health | R01-HL90152 | Asrar B Malik |
| National Institutes of Health | T32-HL007829 | Asrar B Malik |

The funders had no role in study design, data collection, and interpretation, or the decision to submit the work for publication.

### Author contributions

Long Shuang Huang, Conceptualization, Resources, Data curation, Formal analysis, Investigation, Visualization, Methodology, Writing – original draft, Project administration; Mohammad Anas, Conceptualization, Resources, Data curation, Formal analysis, Validation, Investigation, Visualization, Methodology, Writing – original draft, Project administration; Jingsong Xu, Writing – review and editing; Bisheng Zhou, Resources, Methodology; Peter T Toth, Supervision, Investigation, Visualization, Methodology; Yamuna Krishnan, Resources, Supervision, Methodology, Writing – review and editing; Anke Di, Conceptualization, Data curation, Formal analysis, Supervision, Validation,

Investigation, Visualization, Methodology, Writing – original draft, Project administration, Writing – review and editing; Asrar B Malik, Conceptualization, Resources, Supervision, Funding acquisition, Project administration, Writing – review and editing

### Author ORCIDs
Yamuna Krishnan http://orcid.org/0000-0001-5282-8852
Anke Di http://orcid.org/0000-0001-5055-1407
Asrar B Malik http://orcid.org/0000-0002-8205-7128

### Ethics
All mice were housed in the University of Illinois Animal Care Facility in accordance with institutional and NIH guidelines. Veterinary care and animal experiments were approved by the University of Illinois Animal Care & Use Committee (ACC protocol number: 21-032).

### Decision letter and Author response
Decision letter https://doi.org/10.7554/eLife.83842.sa1
Author response https://doi.org/10.7554/eLife.83842.sa2

## Additional files

### Supplementary files
• MDAR checklist

### Data availability
All data generated or analyzed during this study are included in the manuscript. Source Data have been provided for Figures 3–6 for the western blots.

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
