## [Editor Report]

This important study focuses on the mechanism by which potassium channels are activated prior to NLRP3 inflammasome activation. Using convincing microscopy and biochemical studies, the authors demonstrate that the potassium channel, TWIK2, located in the endosomal compartment during basal conditions, is translocated onto the plasmalemma upon ATP stimulation, which triggers potassium efflux and subsequent NLRP3 inflammasome activation. These findings address a long-standing question in the inflammasome field.

---

## [Decision Letter]

**Decision letter after peer review:**

Thank you for submitting your article "Endosomal Trafficking of Two Pore K^+^ Efflux Channel TWIK2 to Plasmalemma Mediates NLRP3 Inflammasome Activation and Inflammatory Injury" for consideration by *eLife*. Your article has been reviewed by 3 peer reviewers, one of whom is a member of our Board of Reviewing Editors, and the evaluation has been overseen by Carla Rothlin as the Senior Editor. The reviewers have opted to remain anonymous.

Essential revision:

1. The authors arrive at their conclusions regarding the role of Rab11a-dependent TWIK2-translocation to the plasmalemma in K^+^ efflux-mediated NLRP3 inflammasome activation using a single stimulus, treatment with extracellular ATP. K^+^ efflux is involved in NLRP3 activation by numerous stimuli including nigericin, pore-forming toxins such as hemolysin, LL-OMe, particulate matter such as silica, and infection with pathogenic bacteria such as pneumococcus. The authors should include many of these activators in addition to ATP in their experiments to conclusively show that Rab11a-TWIK2 translocation- K^+^ axis is a universal requirement for K^+^ efflux-mediated NLRP3 activation.

2. The authors use ip injection of LPS in their in vivo experiment. It is well-established that LPS injection in vivo activates the caspase-11-dependent noncanonical NLRP3 inflammasome. To connect their in vitro and in vivo findings and increase the relevance of their in vivo data, the authors should show that Rab11a-TWIK2 translocation- K^+^ axis is required for noncanonical NLRP3 inflammasome activation in vitro. They could use LPS transfection or infection with Gram negative bacteria such as *E. coli*.

4. Figure 3, 4, 5: in their inhibitor/siRNA experiments the authors should measure control inflammasome-independent cytokines such as IL-6 or TNF to show that the cells are functional, and the defect observed is inflammasome specific.

5. To establish the role of Rab11a, P2X7 and calcium influx in mediating the translocation of TWIK2 to the plasma membrane, the authors would need to perform the following experiments:

– Immunostaining of TWIK2 in wildtype and Rab11a-depleted macrophages following ATP stimulation to compare the distribution of TWIK2.

– Immunostaining of TWIK2 in wildtype and P2X7-KO macrophages following ATP stimulation to compare the distribution of TWIK2.

– Immunostaining of TWIK2 in wildtype macrophages left untreated, treated with vacuolin/BAPTA, and removed calcium, followed by ATP stimulation to compare the distribution of TWIK2.

6. The incomplete characterization of the inflammasome activation can be addressed by:

– Providing IL-1b and IL-18 data to support Figure 3C-F.

– Providing IL-1b and IL-18 data to support Figure 4D-G.

– Providing IL-1b and IL-18 data to support Figure 6B-D.

– Providing caspase-1 western blot data to support Figure 4A-C.

7. The colocalization data in Figure 2 should be quantified. Some co-localization signals between TWIK2 and the ER marker PDI can be seen but this was not addressed in the manuscript.

8. The time course of TWIK2 localization in Figure 1a is unconvincing, as it only appears that the abundance of TWIK2 reduces in the ATP-treated cells relative to the control cells. The use of plasma membrane markers could reinforce this conclusion. These experiments should be performed in BMDMs rather than RAW264.7 cells which are not a good model for inflammasomes given their absence of ASC and altered NF-κB/MAPK signaling. Why was TWIK2-GFP used in immunofluorescence microscopy when an antibody was available? Antibody staining specificity could be validated with TWIK2 knockouts.

9. For the NLRP3 inflammasome activation experiments carried out in Figures 3, 4 and 5, a knockdown or knockout of TWIK2 should be included to reinforce the role of TWIK2 in this activation. As it stands, the results only suggest that something in the Rab11a-containing recycling-endosome is leading to activation of the inflammasome. Authors have not shown that Rab11a deficiency reduces TWIK2 trafficking in Figure 5.

10. Authors conclusions on inflammasome activation defects with Rab11a can only be supported when western blotting and other experiments ruling out effects on pro-IL-1β expression and NLRP3 priming. That Rab11a only acts to prevent TWIK2 trafficking has not been conclusively shown. Vacuolin can inhibit many processes, including autophagy, lysosome exocytosis etc. therefore, its use does not necessarily only implicate Rab11a-mediated TWIK2 trafficking.

11. Would the loss of NLRP3 activation following ca^2+^ depletion, BAPTA treatment or Vacuole treatment be reversed via expression of a plasma-membrane targeting TWIK2 variant (as has been done for other proteins via incorporation of the N-terminal RTK palmitoylation/myristoylation motif)? Say under the control of an inducible promoter to prevent constitutive inflammasome activation?

*Reviewer #1 (Recommendations for the authors):*

This important study by Di et al., focuses on the mechanism by which potassium channels are activated prior to NLRP3 inflammasome activation. Using confocal- and electron-microscopy studies the authors demonstrate that the potassium channel, TWIK2, located in the endosomal compartment during basal conditions, is translocated onto the plasmalemma upon ATP stimulation. The authors suggest that this translocation triggers potassium efflux and subsequent NLRP3 inflammasome activation. Using Rab11a-deficient cells, the authors also show an essential role for Rab11a in this process. This is a well written study that has novel findings that are of interest to the inflammasome field. However, multiple issues need to be addressed and important controls should be included to sufficiently justify the authors' conclusions. Detailed comments are given below.

1. The authors arrive at their conclusions regarding the role of Rab11a-dependent TWIK2-translocation to the plasmalemma in K^+^ efflux-mediated NLRP3 inflammasome activation using a single stimulus, treatment with extracellular ATP. K^+^ efflux is involved in NLRP3 activation by numerous stimuli including nigericin, pore-forming toxins such as hemolysin, LL-OMe, particulate matter such as silica, and infection with pathogenic bacteria such as pneumococcus. The authors should include many of these activators in addition to ATP in their experiments to conclusively show that Rab11a-TWIK2 translocation- K^+^ axis is a universal requirement for K^+^ efflux-mediated NLRP3 activation.

2. The authors use ip injection of LPS in their in vivo experiment. It is well-established that LPS injection in vivo activates the caspase-11-dependent noncanonical NLRP3 inflammasome. To connect their in vitro and in vivo findings and increase the relevance of their in vivo data, the authors should show that Rab11a-TWIK2 translocation- K^+^ axis is required for noncanonical NLRP3 inflammasome activation in vitro. They could use LPS transfection or infection with Gram negative bacteria such as *E. coli*.

3. Figure 3A-B: The authors conclude that ATP-induced exocytic event is required for plasmalemma K^+^ efflux by measuring membrane capacitance increase. Does this specifically measure K^+^ efflux? If not, the authors should directly assess K^+^ efflux by measuring intracellular K^+^ levels via commonly used methods such as (APG)4 and pluronic acid F-127 assays ( Refer Eil et al., Nature 2016; Prindle et al., Nature 2015).

4. Figure 3, 4, 5: in their inhibitor/siRNA experiments the authors should measure control inflammasome-independent cytokines such as IL-6 or TNF to show that the cells are functional, and the defect observed is inflammasome specific.

5. There are discrepancies is the figure call outs in the text. Eg: Figure 5 is called out as Figure 4 in the text and Figure 6 as Figure 5.

6. Authors mention that they evaluated the siRNA-Rab11a efficiency via western blot, however, have not included the data in the manuscript. This western image should be included to show that the siRNA treatment in fact depleted Rab11a (both in the in vitro and in vivo experiments).

7. As mentioned in comment 2, LPS injection activates noncanonical inflammasome in mice. In this pathway, K^+^ efflux-mediated NLRP3 inflammasome activation occurs downstream of gasdermin D activation. To show that the defect they observed in mice that received Rab11a-depleted macrophages is specific to the NLRP3 arm of the pathway and not an upstream event, they should assess gasdermin D cleavage in the lung lysates. Similarly, they should measure inflammasome-independent cytokines such as TNF or IL-6 to show that Rab11a-deficiency does not affect upstream TLR4 activation.

8. The authors should include the number of mice they used for each experiment in the figure legend.

*Reviewer #2 (Recommendations for the authors):*

1. To establish the role of Rab11a, P2X7 and calcium influx in mediating the translocation of TWIK2 to the plasma membrane, the authors would need to perform the following experiments:

– Immunostaining of TWIK2 in wildtype and Rab11a-depleted macrophages following ATP stimulation to compare the distribution of TWIK2.

– Immunostaining of TWIK2 in wildtype and P2X7-KO macrophages following ATP stimulation to compare the distribution of TWIK2.

– Immunostaining of TWIK2 in wildtype macrophages left untreated, treated with vacuolin/BAPTA, and removed calcium, followed by ATP stimulation to compare the distribution of TWIK2.

2. The incomplete characterisation of the inflammasome activation can be addressed by:

– Providing IL-1b and IL-18 data to support Figure 3C-F.

– Providing IL-1b and IL-18 data to support Figure 4D-G.

– Providing IL-1b and IL-18 data to support Figure 6B-D.

– Providing caspase-1 western blot data to support Figure 4A-C.

3. The colocalisation data in Figure 2 should be quantified. Some co-localisation signals between TWIK2 and the ER marker PDI can be seen but this was not addressed in the manuscript.

4. The statistical analysis in Figure 5G is not applied appropriately. The comparison should be made between LPS+ATP (scRNA) versus LPS+ATP (Rab11a siRNA).

5. Please fix labelling errors: The Rab11a section should be referred to Figure 5 not Figure 4. The in vivo data should b

*Reviewer #3 (Recommendations for the authors):*

1. In the introduction, (p5 of 38), "This essential function of K^+^ TWIK2 mediated efflux in initiating macrophage inflammatory response" suggests that TWIK2 is an essential component of all inflammatory responses, whereas the authors' previous Immunity paper shows that it is essential only for the inflammatory response to the extracellular DAMP ATP. Please clarify this in the text.

2. The time course of TWIK2 localisation in Figure 1a is unconvincing, as it only appears that the abundance of TWIK2 reduces in the ATP-treated cells relative to the control cells. The use of plasma membrane markers could reinforce this conclusion. These experiments should be performed in BMDMs rather than RAW264.7 cells which are not a good model for inflammasomes given their absence of ASC and altered NF-κB/MAPK signalling. Why was TWIK2-GFP used in immunofluorescence microscopy when an antibody was available? Antibody staining specificity could be validated with TWIK2 knockouts.

3. For the NLRP3 inflammasome activation experiments carried out in Figures 3, 4 and 5, a knockdown or knockout of TWIK2 should be included to reinforce the role of TWIK2 in this activation. As it stands, the results only suggest that something in the Rab11a-containing recycling-endosome is leading to activation of the inflammasome. Authors have not shown that Rab11a deficiency reduces TWIK2 trafficking in Figure 5.

4. Authors conclusions on inflammasome activation defects with Rab11a can only be supported when western blotting and other experiments ruling out effects on pro-IL-1β expression and NLRP3 priming. That Rab11a only acts to prevent TWIK2 trafficking has not been conclusively shown. Vacuolin can inhibit many processes, including autophagy, lysosome exocytosis etc. therefore, its use does not necessarily only implicate Rab11a-mediated TWIK2 trafficking.

5. Would the loss of NLRP3 activation following ca^2+^ depletion, BAPTA treatment or Vacuole treatment be reversed via expression of a plasma-membrane targeting TWIK2 variant (as has been done for other proteins via incorporation of the N-terminal RTK palmitoylation/myristoylation motif)? Say under the control of an inducible promoter to prevent constitutive inflammasome activation?

---

## [Author Response]

Essential revision:1. The authors arrive at their conclusions regarding the role of Rab11a-dependent TWIK2-translocation to the plasmalemma in K^+^ efflux-mediated NLRP3 inflammasome activation using a single stimulus, treatment with extracellular ATP. K^+^ efflux is involved in NLRP3 activation by numerous stimuli including nigericin, pore-forming toxins such as hemolysin, LL-OMe, particulate matter such as silica, and infection with pathogenic bacteria such as pneumococcus. The authors should include many of these activators in addition to ATP in their experiments to conclusively show that Rab11a-TWIK2 translocation- K^+^ axis is a universal requirement for K^+^ efflux-mediated NLRP3 activation.

These are good suggestions. We do not argue Rab11a-TWIK2 translocation- K^+^ axis is a universal requirement for K^+^ efflux-mediated NLRP3 activation. Also we do not think Rab11a is involved in nigericin mediated changes, it has a role in pore-forming toxins such as hemolysin, LL-OMe, particulate matter such as silica, and in mediating infection with other bacteria than the model we used. This statement is based on findings that TWIK2 does not play a role in pore-forming toxins such as nigericin and infection such as *Salmonella*-induced NLRP3 activation; please see the Figure 3 and related text in our *Immunity* paper ^1^.

2. The authors use ip injection of LPS in their in vivo experiment. It is well-established that LPS injection in vivo activates the caspase-11-dependent noncanonical NLRP3 inflammasome. To connect their in vitro and in vivo findings and increase the relevance of their in vivo data, the authors should show that Rab11a-TWIK2 translocation- K^+^ axis is required for noncanonical NLRP3 inflammasome activation in vitro. They could use LPS transfection or infection with Gram negative bacteria such as *E. coli*.

This is a good insight, but we do not argue based on our in vivo data that Rab11a-TWIK2 translocation- K^+^ axis is in fact the sole pathway for LPS-induced NLRP3 inflammation. There potentially could be other pathways including caspase-11-dependent noncanonical NLRP3 inflammasome in LPS-induced NLRP3 inflammasome activation. Our in vitro data clearly showed that Rab11a-TWIK2 translocation- K^+^ axis is a fundamental pathway mediating LPS-induced NLRP3 inflammasome activation. This was the focus and the important point of this study.

On the other hand, Caspase 11-dependent non-canonical NLRP3 inflammasome activation does not need K^+^ efflux (see review ^2^). Caspase 11 directly activated NLRP3 inflammasome, thus we do not think Rab11a-TWIK2 translocation- K^+^ axis is required for noncanonical NLRP3 inflammasome activation.

4. Figure 3, 4, 5: in their inhibitor/siRNA experiments the authors should measure control inflammasome-independent cytokines such as IL-6 or TNF to show that the cells are functional, and the defect observed is inflammasome specific.

We have added TNF-α as control in these experiments.

5. To establish the role of Rab11a, P2X7 and calcium influx in mediating the translocation of TWIK2 to the plasma membrane, the authors would need to perform the following experiments:– Immunostaining of TWIK2 in wildtype and Rab11a-depleted macrophages following ATP stimulation to compare the distribution of TWIK2.– Immunostaining of TWIK2 in wildtype and P2X7-KO macrophages following ATP stimulation to compare the distribution of TWIK2.– Immunostaining of TWIK2 in wildtype macrophages left untreated, treated with vacuolin/BAPTA, and removed calcium, followed by ATP stimulation to compare the distribution of TWIK2.

We have added these experiments in Figure 5C-F. Since the TWIK2 staining by TWIK2 antibody is weak in mouse monocyte-derived macrophages (MDMs), and also it is hard to transfect MDMs with TWIK2 plasmids, we used the mouse cell line – RAW 264.7 macrophage cells transfected TWIK2-GFP plasmid in these studies.

6. The incomplete characterization of the inflammasome activation can be addressed by:– Providing IL-1b and IL-18 data to support Figure 3C-F.– Providing IL-1b and IL-18 data to support Figure 4D-G.– Providing IL-1b and IL-18 data to support Figure 6B-D.– Providing caspase-1 western blot data to support Figure 4A-C.

We have added IL-1β and IL-18 to the original Figures 3C-F, 4D-G, 6D-D in Figure 3G-H, 4I-J, 6E-F. We have also added the caspase-1 Western blot results to support the original Figure 4A-C in Figure 4G.

7. The colocalization data in Figure 2 should be quantified. Some co-localization signals between TWIK2 and the ER marker PDI can be seen but this was not addressed in the manuscript.

We have quantified these data (Figure 2E).

8. The time course of TWIK2 localization in Figure 1a is unconvincing, as it only appears that the abundance of TWIK2 reduces in the ATP-treated cells relative to the control cells. The use of plasma membrane markers could reinforce this conclusion. These experiments should be performed in BMDMs rather than RAW264.7 cells which are not a good model for inflammasomes given their absence of ASC and altered NF-κB/MAPK signaling. Why was TWIK2-GFP used in immunofluorescence microscopy when an antibody was available? Antibody staining specificity could be validated with TWIK2 knockouts.

First, the reason for “it only appears that the abundance of TWIK2 reduces in the ATP-treated cells relative to control cells (in the time course of TWIK2 localization in Figure 1A)” is that these images were taken from videos in which the resolution was poor, thus the distributed TWIK2 in the plasma membrane was difficult to see. We have now used membrane marker to confirm the TWIK2 plasma membrane translocation after ATP challenge (Figure 1B).

Second, the TWIK2 staining by TWIK2 antibody is weak in mouse monocyte- derived macrophages (thus it is also hard to validate the TWIK2 antibody staining specificity with TWIK2 knockouts MDMs). Furthermore, it’s hard to transfect MDMs with TWIK2 plasmids. Thus in these experiments, we used RAW264.7 cells transfected with TWIK2-GFP. We studied TWIK2 translocation, not NLRP3 inflammasome activation (all NLRP3 inflammasome activation in this study was carried out in MDMs); thus, absence of ASC and altered NF-κB/MAPK signaling should not affect TWIK2 translocation.

9. For the NLRP3 inflammasome activation experiments carried out in Figures 3, 4 and 5, a knockdown or knockout of TWIK2 should be included to reinforce the role of TWIK2 in this activation. As it stands, the results only suggest that something in the Rab11a-containing recycling-endosome is leading to activation of the inflammasome. Authors have not shown that Rab11a deficiency reduces TWIK2 trafficking in Figure 5.

We have previously shown the key role of TWIK2 in the mechanism of NLRP3 inflammasome activation in our *Immunity* paper ^1^. Here we focussed specifically on the role of Rab11a in NLRP3 inflammasome activation. In this context, we show additional new added Rab11a deficiency reduces TWIK2 trafficking in Figure 5C-D.

10. Authors conclusions on inflammasome activation defects with Rab11a can only be supported when western blotting and other experiments ruling out effects on pro-IL-1β expression and NLRP3 priming. That Rab11a only acts to prevent TWIK2 trafficking has not been conclusively shown. Vacuolin can inhibit many processes, including autophagy, lysosome exocytosis etc. therefore, its use does not necessarily only implicate Rab11a-mediated TWIK2 trafficking.

We have added the effects of siRNA-Rab11a on pro-IL-1β and NLRP3 priming using Western blotting (Figure 5 I). The results showed siRNA-depletion of Rab11a treatment did not affect pro-IL-1β expression and NLRP3 priming (Figure 5 L).

Vacuolin was used based on its inhibitory role on both endosome trafficking^3^ and exocytosis of lysosomes ^4^. Our electrophysiological studies (both capacitance measurements as an indicator of exocytosis and whole cell current measurements as an indicator of activation of ion channels) show not only ATP-induced exocytosis was blocked by Vacuolin (Figure 3A); but importantly, they also showed that ATP-induced potassium current was blocked in the cells treated with vacuolin (Figure 3B). These results indicate that ATP-induced TWIK2 activation was the result of the plasmembrane fusion event. This was our intent in using vacuolin here. We don’t argue its use here only implicates Rab11a-mediated TWIK2 trafficking.

On the other hand, the functional study of NLRP3 inflammasome activation induced by ATP was inhibited by vacuolin (Figure 3C-H). Also electrophysiological studies showed that ATP-induced potassium channel activation occurred via the plasmlemma membrane fusion event (Figure 3A-B). These result together provide strong evidence that plasma membrane fusion event was linked to the potassium channel activation; hence, this was our intent in using vacuolin for these studies. The key evidence for Rab11a-mediated TWIK2 trafficking is shown in Figure 5C-D in this revised version of this paper.

11. Would the loss of NLRP3 activation following ca^2+^ depletion, BAPTA treatment or Vacuole treatment be reversed via expression of a plasma-membrane targeting TWIK2 variant (as has been done for other proteins via incorporation of the N-terminal RTK palmitoylation/myristoylation motif)? Say under the control of an inducible promoter to prevent constitutive inflammasome activation?

This is a great question. Currently, we do not have means or are able to make such TWIK2 variants.

References

1. Di, A., et al. The TWIK2 Potassium Efflux Channel in Macrophages Mediates NLRP3 Inflammasome-Induced Inflammation. *Immunity* 49, 56-65 e54 (2018).

2. Man, S.M. & Kanneganti, T.D. Regulation of inflammasome activation. *Immunol Rev* 265, 6-21 (2015).

3. Ye, Z.*, et al.* Vacuolin-1 inhibits endosomal trafficking and metastasis via CapZbeta. *Oncogene* 40, 1775-1791 (2021).

4. Cerny, J.*, et al.* The small chemical vacuolin-1 inhibits Ca(2+)-dependent lysosomal exocytosis but not cell resealing. *EMBO Rep* 5, 883-888 (2004).